# Structural Insights into the Evolutionarily Conserved BAF Chromatin Remodeling Complex

**DOI:** 10.3390/biology9070146

**Published:** 2020-06-30

**Authors:** Ryan D. Marcum, Alexis A. Reyes, Yuan He

**Affiliations:** 1Department of Molecular Biosciences, Northwestern University, 2205 Tech Drive, Evanston, IL 60208-3500, USA; ryanmarcum@u.northwestern.edu (R.D.M.); alexisreyes2021@u.northwestern.edu (A.A.R.); 2Interdisciplinary Biological Sciences Program, Northwestern University, 2205 Tech Drive, Evanston, IL 60208-3500, USA; 3Robert H. Lurie Comprehensive Cancer Center of Northwestern University, Northwestern University, 676 N. St. Clair, Chicago, IL 60611, USA

**Keywords:** chromatin remodeling, cancer, SWI/SNF complex, BAF complex, cryo-EM

## Abstract

The switch/sucrose nonfermentable (SWI/SNF) family of proteins acts to regulate chromatin accessibility and plays an essential role in multiple cellular processes. A high frequency of mutations has been found in SWI/SNF family subunits by exome sequencing in human cancer, and multiple studies support its role in tumor suppression. Recent structural studies of yeast SWI/SNF and its human homolog, BAF (BRG1/BRM associated factor), have provided a model for their complex assembly and their interaction with nucleosomal substrates, revealing the molecular function of individual subunits as well as the potential impact of cancer-associated mutations on the remodeling function. Here we review the structural conservation between yeast SWI/SNF and BAF and examine the role of highly mutated subunits within the BAF complex.

## 1. Introduction

A primary means of maintaining genomic stability and tuning gene expression in eukaryotes is the formation of chromatin and regulation of chromatin architecture. Eukaryotes rely on a variety of ATP-dependent chromatin remodelers to regulate the movement, spacing, and composition of nucleosomes [1,2]. Some of the most abundant chromatin remodelers belong to the switch/sucrose nonfermentable (SWI/SNF) family. The SWI/SNF family remodelers act in many essential cellular processes through their ability to move and remove nucleosomes, regulating the accessibility of DNA to promote processes such as transcription initiation and DNA repair. The essential role of the SWI/SNF family of remodelers is evident in their conservation from yeast (SWI/SNF and Remodel the Structure of Chromatin or RSC) to humans (BAF and PBAF) and the critical effect of their mutations on organism survival and development.

Over the last few decades, a series of studies has identified mutations in multiple SWI/SNF subunits within a variety of tumor types, and whole-genome sequencing of cancerous primary tumors revealed that the human BAF complex exhibited a high mutation frequency (~20%) [3,4,5]. For this reason, SWI/SNF subunit mutations have become popular targets for potential biomarkers and cancer therapeutics [5,6]. Mutations in the BAF complex were broadly distributed among tumor types compared to other known tumor suppressors with the exception of TP53 [3,7]. This suggests that BAF complex activity can contribute to general tumor suppression and its mutations facilitate disease progression. It is also likely that these mutations can alter the function of SWI/SNF complexes, allowing for dysregulation of chromatin architecture in cancers. The most frequently mutated subunits are SMARCA2/4 (4.5%) and AT-rich interaction domain 1A/1B (ARID1A/B) (14.9%) [3]. Recent high-resolution structures of both yeast SWI/SNF (ySWI/SNF) and human BAF (hBAF) complexes provide further mechanistic understanding of the remodeler function when bound to nucleosomes [8,9], allowing us to place mutations from these key subunits in context of a 3D model of the full complex. Here we review the newly determined structures of BAF family complexes and the role of commonly mutated subunits and discuss the functional effects that these mutations have on SWI/SNF chromatin remodelers.

## 2. Architecture of the BAF Family Remodeler

The BAF complex is composed of multiple subunits, some of which exist as different isoforms, allowing for tissue-specific regulation. With the recent publications of multiple SWI/SNF complex structures, the various subunits can be organized into three modules: the adenosine triphosphatase (ATPase), the actin-related protein (Arp), and the Body modules [8,9,10,11].

### 2.1. ATPase Module

As the name indicates, the ATPase module harbors the ATPase subunit that in ySWI/SNF is Snf2/Swi2 and in hBAF is SMARCA4/BRG1/BAF190A. The ATPase module occupies a major part of the catalytic subunit for all ATP-dependent chromatin remodelers and contains RecA-like domains that are responsible for coupling ATP hydrolysis to DNA translocation (Figure 1). The ATPase module of ySWI/SNF and hBAF makes extensive contacts with nucleosomal DNA, centered near superhelical location (SHL) +2 of the nucleosome. SHL +2 has previously been established as the binding site for Snf2/Swi2, allowing for the initial distortion of DNA for remodeling [12,13,14]. The hBAF structure solved by He et al. reveals the location of the two RecA-like ATPase lobes of SMARCA4 to be centered at SHL +2.5 [9], suggesting an open pre-engaged state in the absence of ATP. In contrast, the RecA-like lobes of the Snf2/Swi2 structure from Han et al. show the closed, engaged state at SHL +2 in the presence of an ATP analog [8]. These conformational differences suggest a model by which the ATPase subunits of SWI/SNF complexes closely engage the nucleosome in the presence of ATP, allowing for DNA binding and translocation after transitioning from an open state in the absence of ATP. It is worth noting that for both complexes, the C-terminal bromodomains, which are known to interact with acetylated lysine on histone tails and believed to play a regulatory role in chromatin remodeling, are unresolved [15,16,17,18,19]. In addition to its catalytic activity [20,21], SMARCA4 has also been shown to play a role in localization of the BAF complex to genomic targets [22] and eviction of PRC1 (polycomb repressive complex 1) from chromatin [23]. This emphasizes the variety of functions in which SMARCA4 is involved.

Previous studies highlighted the important role of SMARCA4 as a tumor suppressor. It has been observed that cancer cell lines are deficient in SMARCA4 and that ectopic expression of SMARCA4 can rescue cell-cycle inhibition [24,25,26]. Null mutations of SMARCA4 lead to death during embryogenesis in mouse models; however, mice with heterozygous mutations of SMARCA4 were instead predisposed to tumor development [27,28]. Genetic studies have revealed high frequencies of SMARCA4 mutations in ovarian [29,30,31], thoracic [32], lung [33], and prostate [34] cancer.

Although cancer-associated mutations are present throughout SMARCA4, mutations to the two ATPase lobes are commonly found in cancer [35]. Immunoprecipitation studies showed that cancer-associated mutations to the SMARCA4 ATPase domain do not prevent its incorporation into the BAF complex [36]. However, the engagement and release of chromatin as well as PRC1 eviction are disrupted by cancer-associated mutations to the ATP recognition cleft or DNA-binding groove of the ATPase domain [23,35].

In addition to its catalytic role, SMARCA4 is an important scaffolding protein in the BAF complex. The N-terminal helicase-SANT-associated (HSA) domains of both Snf2/Swi2 and SMARCA4 extend into the other modules and play critical roles in bridging the ATPase and Body modules of their respective complexes.

### 2.2. Arp Module

The SWI/SNF Arp module functions as a bridge between the ATPase and Body modules (Figure 1). The ySWI/SNF Arp module consists of the Snf2 HSA domain, Arp7, and Arp9, and in the hBAF complex, the Arp module consists of the SMARCA4 HSA domain, ACTB/β-actin, and ACTL6A/BAF53A. For both ySWI/SNF and hBAF, the Arp module is connected to the ATPase and Body modules through the single α-helix of the HSA domains, which is necessary to maintain the sandwiched placement of the nucleosome relative to the full complex. The HSA domain extends through the Arp module into the Body module, acting as an anchor for the rest of the complex.

Knockout of SMARCA4 did not disrupt formation of the Body module in coimmunoprecipitation or biochemical fraction experiments [37]. However, deletion of the HSA was shown to disrupt the interaction between SMARCA4 and ARID1A [38], indicating that other components of the Arp module are insufficient to bridge the interaction between the Body and ATPase modules. The actin-like proteins within the Arp module may instead function to add stability to the flexible HSA domain and play regulatory roles. The mechanism of Arp module regulation of ATPase subunits remains unclear, but biochemical evidence shows that the actin-like proteins within the Arp module regulate primary functions such as DNA binding, ATPase activity, and DNA translocation [39,40]. The actin-like proteins also regulate secondary functions of SWI/SNF chromatin remodelers, such as their role in cellular differentiation seen in the switch from ACTL6A/BAF53A to ACTL6B/BAF53B during neuronal development [41]. In silico analysis of cancer-associated mutations of SMARCA4 suggest that they destabilize the HSA α-helix [42]. This suggests another mechanism of BAF complex inactivation in cancer wherein truncating or point mutations to the HSA domain can uncouple the Body module from the ATPase module.

### 2.3. Body Module

The Body module is packed against the nucleosome face opposite from the ATPase module and assists in maintaining complex integrity and histone octamer immobilization. The Body module of ySWI/SNF consists of Swi1, Snf5/Swi10, Swi3, Snf12/Swp73, Snf6, and Swp82. In the hBAF complex, the Body module consists of ARID1A/BAF250A, SMARCB1/BAF47/hSNF5, SMARCC2A/BAF170, SMARCC2B, SMARCD1/BAF60A, SMARCE1/BAF57, and DPF2/BAF45D. SMARCE1 and DPF2 are both specific to hBAF and do not have any orthologs in ySWI/SNF. However, a comparison of the yeast and human structures showed that Snf6 and Swp82 share the same approximate position within the complex as SMARCE1 and DPF2, respectively, and potentially fulfill the same function within their respective complexes (Figure 1).

A vast majority of SWI/SNF complex mass comes from the Body modules, which require an intricate network of interactions to maintain complex stability [8,9]. Within the SWI/SNF Body module, Swi1 (yeast) or ARID1A (human) acts as hub for the majority of the stabilizing interactions. Han et al. showed that Swi1 has a vast interface of connections with other Body subunits, including Snf5/Swi10, Swi3, and Snf12/Swp73 [8]. Similarly, the nucleosome-bound hBAF complex shows the integral nature of ARID1A in the complex, with interactions occurring between the core of ARID1A and nearly every subunit of the Body module [9]. As mentioned previously, ARID1A also bridges the interaction between the Arp and Body modules, demonstrating the important role of ARID1A as a scaffold for the assembly of the hBAF complex.

Mutant BAF complexes were evaluated by biochemical fractionation of purified endogenous complexes [37], and it was found that complex stability was compromised by the deletion of ARID1A. Additionally, the Y2254^*^ mutant of ARID1A disrupted the interaction between ARID1A and BAF complex subunits [37]. This truncation mutant removed a two-helix bundle that interacts with SMARCD1, another key scaffolding protein, which may explain the drastic effect on complex stability. Mutations to the C-terminus of ARID1A (including Y2254), which acts as the core of the Body module, are common in cancer [43] (Figure 1) and would be expected to have equally deleterious effects on complex stability.

The role of ARID1A as a scaffold for BAF complex assembly is especially pertinent as there is a high occurrence of inactivating frameshift mutations in ARID1A in cancer [44,45]. Additionally, frame insertions and deletions that disrupt the C-terminal nuclear export signal lead to increased ubiquitination and degradation of ARID1A by the proteasome [46]. Point mutations to G2087, one of the most common cancer-associated mutations in hBAF, also increase ubiquitination and decrease protein levels when expressed in human cell lines [37]. These results demonstrate a common theme of ARID1A depletion in cancer. This is supported by additional studies of cancer cell lines that showed that partial depletion of ARID1A enhanced cell proliferation [47,48] and restoring expression of ARID1A in both ARID1A-deficient breast cancer cell lines and mouse ovarian cancer models was sufficient to inhibit cell proliferation [48,49].

The ARID (AT-rich interaction domain) of ARID1A is known to interact with DNA, and mutations to this domain affect nucleosome-binding affinity [50]. However, as the ARID domain is unresolved, it is not known what role the ARID domain plays in nucleosome remodeling, nor does the current structural model explain how the interaction with DNA could occur. Another unexplored aspect of ARID1A is the mechanism by which it recruits other chromatin-modifying complexes, such as histone deacetylase and coactivator complexes [51]. This highlights the possibility of cancer-associated mutations affecting not just SWI/SNF activity but also recruitment of chromatin-modifying complexes.

The Body module also plays an important role in nucleosome recognition through the Snf5/Swi10 (yeast) or SMARCB1 (human) subunit. SMARCB1 is integrated into the complex through its interactions with ARID1A and other subunits of the Body module, such as SMARCC2 and DPF2. A conserved arginine-rich region within the C-terminus of SMARCB1 is known to be crucial for binding the acidic patch of the nucleosome and coupling ATP hydrolysis and DNA translocation [52].

SMARCB1 demonstrated a low frequency of mutations to coding regions (0.6% of tumors screened) compared to other subunits of the BAF complex [3]. However, loss of SMARCB1 is reported in various cancers including malignant rhabdoid tumors [53,54], epithelioid sarcoma [55,56], medullary carcinoma [57,58], synovial sarcoma [59], familial schwannomatosis [60], extraskeletal myxoid chondrosarcoma [61], and small-cell hepatoblastoma [62]. In addition, loss of SMARCB1 is the defining genetic change in both synovial sarcoma and malignant rhabdoid tumors (MRTs) [63,64].

Synovial sarcoma is characterized by a specific t(X; 18) chromosomal translocation resulting in the SS18-SSX fusion gene [65,66]. This fusion is a defining feature of synovial sarcoma and is diagnostic of this soft-tissue malignancy [67]. Biochemical fractionation studies have demonstrated that this fusion protein displaces SMARCB1 while the rest of the BAF complex remains intact and the SS18-SSX complex is relocalized to oncogenic loci, such as SOX210,11 [68]. This prevents H3K27 methylation by PRC2 (polycomb repressive complex 2) at these loci and leads to gene activation. It is hypothesized that novel interactions with the fusion protein are responsible for relocalization of the mutant BAF complex; however, the mechanism by which this occurs is currently undefined.

SMARCB1 depletion in MRTs leads to a similar loss of PRC2 antagonism [69,70]. However, the mechanism by which this occurs is different. Mutations in MRTs consist of inactivating deletions or truncations of SMARCB1 [53], and loss of SMARCB1 leads to a global decrease in its occupancy at enhancer and promoter regions [71] as opposed to the relocalization seen in synovial sarcoma. While different mechanisms are at play with synovial sarcoma and MRTs, it demonstrates the important role that SMARCB1 plays in both polycomb antagonism and interactions with genomic targets.

As mentioned previously, the C-terminal helix of SMARCB1 is the main contact between the Body module of the BAF complex and the histone octamer through the H2A/H2B acidic patch [9,52]. While loss of SMARCB1 does not affect the assembly of other modules [37], it does negatively affect chromatin remodeling. As noted above, SMARCB1 is required for complex occupancy at enhancer and promoter regions [71]. Additionally, mutations to the C-terminal helix of SMARCB1 disrupt nucleosome binding and chromatin remodeling by the BAF complex [52]. These studies support the critical function of SMARCB1 in facilitating DNA translocation around the histone octamer and the potential disruption of chromatin remodeling by cancer-associated mutations.

## 3. Future Directions

It has been well established that mutations to BAF complex subunits are associated with the progression of multiple cancers. Recent additions of high-resolution models of ySWI/SNF and hBAF complexes have provided insight into the molecular roles of individual subunits and can be used to infer the effect that cancer-associated mutations would have on the integrity and activity of the BAF complex. However, functionally important domains such as ARID in ARID1A and the C-terminal bromodomain of SMARCA4 are unresolved in these structures, leaving their molecular function and role in disease progression unclear. Furthermore, the mechanism of BAF complex relocalization and structural rearrangements in mutant complexes (such as SS18-SSX fusion in synovial sarcoma) are currently unclear. Further investigation is required to understand these important features of the BAF remodeling complex that can provide additional clues into its role in cancer.

## Figures and Tables

**Figure 1 biology-09-00146-f001:**
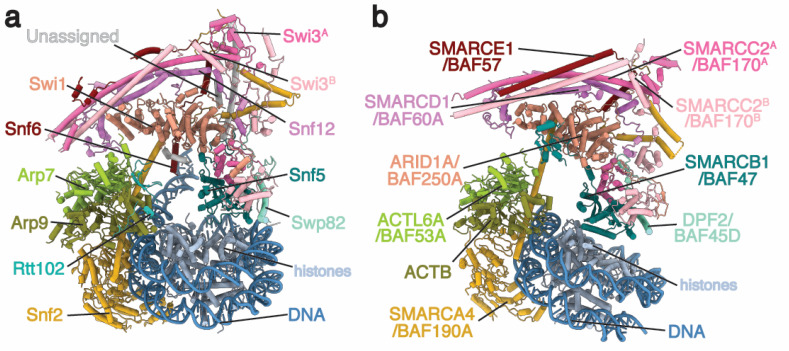
Comparison of switch/sucrose nonfermentable (ySWI/SNF) and human BAF (hBAF) complexes. Structural models of (**a**) ySWI/SNF (PDB 6UXW) and (**b**) hBAF (PDB 6LTJ) are shown in a similar orientation by superimposing orthologous subunits in a pairwise manner. Orthologs are colored the same between models. “/” separates alternative names for individual subunits in hBAF.

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
