# Peer review of "Structural Insights into the Evolutionarily Conserved BAF Chromatin Remodeling Complex"

_biology, 2020, doi:10.3390/biology9070146_

Round 1

Reviewer 1 Report

The authors are reviewing structural data on the BAF chromatin remodeling complex in yeast and human by bringing together a very large collection of papers. Interestingly, they examined in detail the relationship between cancer associated mutations and the structure of the complex. Their analysis allows to apprehend the structure of complex and its important parts in a very comprehensive way.

Minor comment

-I think that a tiny piece of the SMARC4 protein is missing on Figure 1. The protein appears to be in 2 parts. It is likely a copy-paste issue.

Author Response

The Authors would like to thank the reviewers for their comments. Figure 1 has been modified to include a dashed line between the two segments of SMARCA4 that are connected by an unresolved linker.

Reviewer 2 Report

In the mini review "Structural insights into the evolutionarily conserved BAF chromatin remodeling complex", the authors review the structural conservation between yeast SWI/SNF complex and its human homolog BAF complex and examine the role of highly mutated subunits within the BAF complex.

The mini review mainly focuses on the three modules of SWI/SNF complex: the adenosine triphosphatase (ATPase), the actin-related protein (Arp), and the Body modules.

The authors have done a good job of reviewing the the newly determined structures of BAF family complexes and the role of commonly mutated subunits, and discussing the functional effects that these mutations have on SWI/SNF chromatin remodelers. However, a few points need to be addressed to improve the manuscript:

1, Line 59, the sentence "the ATPase module harbors the ATPase subunit that in ySWI/SNF is Snf2/Swi2, whereas in hBAF, it is SMARCA4/BRG1/BAF190A" seems strange to me. The authors may change it to "harboring the ATPase subunit in".

2, Line 83, it is better to change "in ovarian[29-31], thoracic[32] , lung[33], prostate[34] cancer" to "in ovarian[29-31], thoracic[32] , lung[33], and prostate[34] cancer."

3, Line 126-129, the references are needed.

Author Response

The Authors would like to thank the reviewers for their comments.

Lines 57-59 have been changed as suggested for clarity.

Line 80 has been updated to use proper grammar.

Citations have been added for lines 126-129.